# Hybrid Rice Production: A Worldwide Review of Floral Traits and Breeding Technology, with Special Emphasis on China

**DOI:** 10.3390/plants13050578

**Published:** 2024-02-21

**Authors:** Humera Ashraf, Fozia Ghouri, Faheem Shehzad Baloch, Muhammad Azhar Nadeem, Xuelin Fu, Muhammad Qasim Shahid

**Affiliations:** 1State Key Laboratory for Conservation and Utilization of Subtropical Agro-Bioresources, South China Agricultural University, Guangzhou 510642, China; humeraa49@gmail.com (H.A.); foziaghouri@scau.edu.cn (F.G.); fuxuelin@scau.edu.cn (X.F.); 2Guangdong Provincial Key Laboratory of Plant Molecular Breeding, College of Agriculture, South China Agricultural University, Guangzhou 510642, China; 3Department of Biotechnology, Faculty of Science, Mersin University, Mersin 33100, Türkiye; balochfaheem13@gmail.com; 4Faculty of Agricultural Sciences and Technologies, Sivas University of Science and Technology, Sivas 58140, Türkiye; azharjoiya22@gmail.com

**Keywords:** rice breeding, genomic approaches, floret, flower, hybrid rice

## Abstract

Rice is an important diet source for the majority of the world’s population, and meeting the growing need for rice requires significant improvements at the production level. Hybrid rice production has been a significant breakthrough in this regard, and the floral traits play a major role in the development of hybrid rice. In grass species, rice has structural units called florets and spikelets and contains different floret organs such as lemma, palea, style length, anther, and stigma exsertion. These floral organs are crucial in enhancing rice production and uplifting rice cultivation at a broader level. Recent advances in breeding techniques also provide knowledge about different floral organs and how they can be improved by using biotechnological techniques for better production of rice. The rice flower holds immense significance and is the primary focal point for researchers working on rice molecular biology. Furthermore, the unique genetics of rice play a significant role in maintaining its floral structure. However, to improve rice varieties further, we need to identify the genomic regions through mapping of QTLs (quantitative trait loci) or by using GWAS (genome-wide association studies) and their validation should be performed by developing user-friendly molecular markers, such as Kompetitive allele-specific PCR (KASP). This review outlines the role of different floral traits and the benefits of using modern biotechnological approaches to improve hybrid rice production. It focuses on how floral traits are interrelated and their possible contribution to hybrid rice production to satisfy future rice demand. We discuss the significance of different floral traits, techniques, and breeding approaches in hybrid rice production. We provide a historical perspective of hybrid rice production and its current status and outline the challenges and opportunities in this field.

## 1. Introduction

Rice is an important source of food, and more than 50% percent of people rely on rice to fulfill their hunger needs globally [1]. Rice production and consumption is perhaps the most critical economic activity, and it is believed that nearly every day, rice food is taken by half of the population globally at least once [2]. Moreover, hybrid rice is important for improving food security and enhancing crop production. Keeping this in view, more work is required to increase its productivity to combat the food scarcity issue [3]. Furthermore, some hybrid types have demonstrated diminished resilience to pests and diseases; however, continued research can create hybrids with enhanced resistance and increased yields. Hybrid rice is a significant breakthrough in modern agriculture [4,5,6,7,8,9,10,11]. In 1972, Professor Yuan Longping developed the Er-jiu-nan-1 A and B lines, which represent the first pairing of male sterile and maintainer lines with the help of his colleagues. Later, the restorer line was found to conclude the “three-line” production system to develop Nanyou 2, the first high-yielding commercial rice hybrid [12]. In China, hybrid technology for rice has been fruitfully implemented for 40 years to produce commercial rice. Because of its high yield and broad adaptability to various climatic conditions, Shanyou 63, a mega hybrid rice, developed and proved to be a turning point for hybrid rice production in China, deriving from two parents such as Minghui 63 and Zhenshan 97A. During the 29 years between 1984 and 2012, mega hybrid seeds were grown in sixteen different divisions of China on approximately seventeen percent of the fields each year. Additionally, the hybrid rice and its parents have been extensively used for fundamental and applied research on stress tolerance, genome research, biological markers, and rice hybrid vigor [13]. Its popularity has grown significantly, as China has become the biggest country in terms of the consumption and production of hybrid rice [14]. Hybrid rice is produced by crossing the male parent (restorer lines) and female parental lines (cytoplasmic male sterile lines (CMS)) [15]. The discovery of CMS lines enabled the researchers to develop more seeds for hybrid rice production, which is key to successful hybrid rice breeding [16]. China invented the hybrid rice variety for the first time in the 1970s by combining a restorer line with a sterile line of the male parent [17].

However, being a self-fertilized crop, the production of hybrid seeds in rice is not an easy step [18]. The development and utilization of floral traits and breeding technologies are crucial in hybrid rice production [19]. For hybrid rice to be successful, floral characteristics like heterosis, outcrossing mechanisms, and male sterility are all important traits at the production level of crops. To improve the hybrid seed set in rice, the genetic variety of floral traits is crucial; among them, anther length is a useful characteristic for hybrid seed production [20]. Therefore, improvement in the efficiency of hybrid seed production is needed, as it is an essential step for sustainable hybrid rice production [21], and the evaluation of floral traits plays an important role in it [22]. Certain floral traits, such as stigma exsertion, spikelet opening angle, flowering behavior, and pollen longevity, can affect the effectiveness of cross pollination [20]. Among these traits, stigma exsertion is particularly effective in increasing cross pollination [23].

In China, a “third-generation” breeding system has been implemented for the development of hybrid seeds. Historically, hybrid rice has been a primary focus of Chinese rice research [24]. Currently, hybrid rice is being developed and used in many countries worldwide, including in sub-Saharan Africa, Egypt, India, Brazil, Bangladesh, Indonesia, the Philippines, Myanmar, Sri Lanka, and some other developing organizations (Table 1). Moreover, scientists are using web-based technologies to pinpoint the ideal climatic conditions to obtain higher yields [24]. The challenges in rice production forced the researcher to rethink the present breeding methods to fulfill the future demands of rice [25]. Moreover, modern breeding technologies can deliver quick and better varieties of rice in a short time. The improvement in the production of hybrid rice is the result of joining certain factors such as advances in breeding technologies, improved understanding of floral traits and genetic mechanisms, and better agronomic practices [26,27,28].

This review aims to explore the evolving landscape for hybrid rice growth and development by examining floral traits, characteristics, and breeding technology in developing new and improved varieties of hybrid rice and the current state of its production, including challenges faced in the breeding process and the potential solutions to overcome them. We highlighted the role of various floral traits, such as male and female sterility, flowering synchrony, pollen viability, and breeding technologies such as molecular markers, gene editing, and transgenic technology, to develop high-yielding hybrid rice varieties. The aim is to give an extensive understanding of the present state of hybrid rice production to identify future research and development directions in this field. The knowledge about different floral traits of rice, their importance in the production of high-yield hybrid rice, and their link to boosting the yield through salient characteristics of hybrid rice are crucial for farmers and researchers alike.

## 2. Challenges Faced in Hybrid Rice Production

The development of hybrid rice has been essential in supplying the growing global demand for food [33]. Various floral traits, i.e., panicle length, grain size, and flowering time act as key determinants of yield and quality in hybrid rice production. The development of panicles and stigma exertion is important for hybrid seed growth because these are essential for seed setting and outcrossing rate. Moreover, sterile stability and better growth of floral characteristics play a key role in successful hybrid seed production. Another important consideration is flowering synchronization since improper synchronization might result in the absence of, or a poor, seed set. In addition, the impact of other environmental factors, such as a shortage of water and its impact on floral traits, should be taken into consideration during the production of seeds [34,35,36,37,38]. However, the improvement of floral traits is challenging due to their complex genetic regulation and interactions with environmental factors [39]. Therefore, exploring the challenges and their importance in floral traits, and, ultimately, in hybrid rice production, is crucial to ensure the sustainable and efficient development of hybrid rice varieties.

This section will discuss the obstacles and their roles in hybrid rice production.

Despite the significant progress made in hybrid rice production, challenges still need to be addressed to improve its yield potential and commercial viability further. Some of these challenges include the following:Maintaining genetic purity: Hybrid rice breeding requires strict maintenance of genetic purity ensuring that the desired traits are consistently passed on to the next generation. Any contamination can result in reduced yield and poor performance of the hybrid [40].High cost of hybrid seeds: Hybrid seeds are more expensive than traditional varieties, which can make them unaffordable for small-scale farmers [29].Inconsistent performance: Hybrid rice can exhibit variability in performance due to genotype-by-environment interactions, making it difficult to predict yield under different growing conditions [41].Limited genetic diversity: Hybrid rice relies on a limited number of parental lines, which can lead to decreased genetic diversity and increased susceptibility to disease and pests [42].Difficulty in achieving high seed purity: The maintenance of seed purity in hybrid rice production is difficult and requires meticulous management practices [43].Lack of infrastructure: Lack of infrastructure for hybrid seed production and distribution can limit the adoption and availability of hybrid rice varieties [4].Limited understanding of genetic mechanisms: Despite advances in genetic research, knowledge of the fundamental genetic pathways that contribute to hybrid vigor in rice is still lacking [44].Poor adaptation to different environments: Hybrid rice varieties may perform well in some environments but may not be well adapted to others, which can limit their widespread adoption [45].The limited availability of high-quality hybrid parents is critical for the success of hybrid rice breeding, but the limited availability of such parents can be a bottleneck in the process [46].Regulatory issues: Regulations related to seed certification, intellectual property rights, and biosafety can pose challenges to the commercialization and adoption of hybrid rice varieties [47].

## 3. Breeding Techniques for Producing Hybrid Rice

Rice breeding is challenging and time consuming, with increasing pressure to produce more rice to nourish the rapidly growing global population [48]. Novel technologies have developed new cultivars in a short time, but land degradation, environmental pollution, and land scarcity continue to pose obstacles [25,48]. To move beyond these obstacles, Rapid Generation Advance (RGA), a new breeding method, has been adopted for inbred rice varieties, which involves planting high-density breeding populations in a greenhouse to promote early flowering and seed setting, resulting in fast generation advancement and reduced costs [49] (Figure 1). Moreover, unique breeding tactics can increase the expression of floral traits, such as stigma exsertion, which plays a crucial role in hybrid rice production [50,51]. Hybrid rice breeding technology has undergone significant advancements over the past few decades. This section discussed the traditional breeding methods for producing hybrid rice, modern breeding, recent advances, molecular markers, genomics approaches, and the development of new hybrid rice breeding strategies (Figure 1).

### 3.1. Contribution of Classical Breeding Approaches in the Production of Hybrid Rice

The process of developing hybrid rice by conventional breeding approaches involves manually cross-pollinating two or more inbred rice lines that possess desirable qualities (Figure 1), which results in the production of hybrid offspring with superior attributes [30]. These inbred lines are typically developed through several generations of self-pollination to create genetically stable lines that breed true [52]. The resulting hybrid offspring often exhibit heterosis or hybrid vigor [53]. In hybrid rice breeding, hybrid vigor or the superiority of the F_1_ hybrids produced by a cross of different parents is used to achieve the desired characteristics. The way the F_1_ hybrid performs and the effectiveness of the hybridization procedure can both be influenced by the floral traits of the parents used for the crossing [20]. Moreover, the floral traits that were discovered to be strongly associated with the outcrossing rate serve as an important selection criterion to develop new and promising parental lines, as well as hybrids, in addition to finding economically viable CMS lines [54]. In the hybrid rice breeding programs, CMS lines served as female parents. They are produced by inserting a CMS gene into the female parent [55]. Improved floral traits, including longer stigma, wider floret opening angle, and higher stigma exsertion, considerably increase the outcrossing of CMS lines [56]. The flowery characteristics of cytoplasmic male sterile lines are crucial components of the hybrid breeding program [57]. However, manual cross pollination is a labor and time-intensive operation, making it difficult to produce large quantities of hybrid seeds for commercial cultivation [58].

### 3.2. Advances in Breeding Technologies for Hybrid Rice

Advances in hybrid rice breeding technology have led to the innovation of new methods that can streamline hybrid seed production. The adoption of RGA and other emerging breeding technologies can drastically cut expenses in rice production, increase efficiency, and accelerate the production of novel varieties [59]. Improvement and identification of key floral traits, such as stigma exsertion through breeding tactics, also offer opportunities to enhance the yield and excellence of hybrid rice varieties [60]. However, it is important to ensure that these technologies are deployed responsibly and sustainably, taking into account ethical and societal implications [61]. Despite these obstacles, rice production’s ongoing investigation of floral features and breeding technologies will be essential to satisfy the world’s growing population [62]. One such technique is the use of male sterile lines, which eliminates the need for manual emasculation and reduces the time and labor required for hybrid seed production [63]. Furthermore, cytoplasmic male sterility, which involves the use of GM (Genetically Modified) cytoplasm to develop sterile male plants, and two-line and three-line hybrid systems to produce hybrids with desired traits by using male sterile lines [64] (Figure 1). Moreover, the generation of hybrid rice seeds involves a two-line hybrid technique, wherein two genetically distinct lines are used due to the self-pollinating nature of the rice crop, and this approach has been devised to address the challenges connected with producing hybrid seeds for rice [29,65].

The male sterile line (A line) and the maintainer line (B line) are initially crossed in the two-line system to produce seed hybrids. The resulting F1 hybrid plants are vigorous hybrids with excellent characteristics from both parental lines [66]. The male sterile line, on the other hand, is unable to produce viable pollen; hence, it is unable to self-pollinate or create seeds. As a result, the hybrid plants do not provide the hybrid seeds directly [67] (Figure 1). In addition, another method for the production of hybrid seeds is the three-line hybrid system (Figure 1), which uses the male sterile line (A line), the maintainer line (B line), and the restorer line (R line), for the production of hybrid seeds. Furthermore, this technique may be used to manufacture hybrid rice seeds on a larger scale [29,68].

Through the application of advanced precise phenotypic methods and modern biotechnological approaches such as QTL mapping and genome-wide association studies (GWAS), significant progress has been achieved in discovering chromosomal regions that are appropriate for enabling direct and indirect selection of linked hybrid potential characteristics [69]. Therefore, it has been demonstrated that anther extrusion and pollen shedding are useful for assessing associated traits such as pollen mass and floral opening duration [70]. There is always a possibility that the superior floral and agronomic trait combinations in potential male and female parents may not pass on desired features to their hybrid offspring. Combining ability has been frequently used in plant breeding to compare the performance of lines in hybrid combinations to address this problem [71]. Moreover, synthetic biology, gene editing, and other techniques are used in the field of molecular breeding design to alter DNA sequences directly. This method simplifies and enhances the breeding process [72]. Moreover, exogenous cyanobacteria applications such as *Nostoc muscorum* and *Anabaena oryzae* enhance the outcrossing rates and floral traits, as well as the production of seeds for hybrid rice [73].

### 3.3. Use of Genomics and Genetic Markers in the Development of Hybrid Rice

Genomics and molecular markers have made a substantial contribution to the development of hybrid rice breeding technology. These tools enable breeders to identify desirable traits more quickly and accurately [74], select parents for hybridization, and track the inheritance of specific traits in hybrid offspring [29]. The use of genomic information has also enabled breeders to develop molecular markers for marker-assisted selection, which can accelerate the breeding process by allowing breeders to select desirable traits with greater accuracy and efficiency [75] (Figure 1). Moreover, by contributing to the discovery, development, and selection of desirable floral characters, molecular markers contribute significantly to the development of hybrid rice [76]. To locate rice genomic regions linked to floral traits, investigation of precise molecular markers is crucial. Researchers can identify chromosomal regions that are significantly associated with important floral traits like stigma length and out-crossing rate by screening genotype mapping population and DNA markers. Using these genomic regions in the breeding system to enhance certain traits in rice can ultimately result in increased hybrid rice production [77]. According to earlier research, a genetic linkage map was developed using 92 SSR markers across the entire genome, and the percentage of stigma exertion was examined; three QTLs were found on different chromosomal numbers [78]. For enhancing the stigma length of rice for hybrid rice production by using marker-assisted selection, the gene-specific LQ30 InDel marker was developed [79]. Molecular markers are employed in marker-assisted selection (MAS) to pinpoint desirable features in parental lines (Figure 1). Breeders can rapidly and reliably test a large number of prospective parental lines by connecting particular markers with qualities of interest [80]. These markers decrease the time and expenses involved in the traditional method of finding the gene of interest [81]. Genomic selection uses genomic data, such as genotypic and molecular information, to determine an individual’s breeding value and pick the best candidates for hybridization. GS models can predict how well hybrid offspring will perform and direct the choice of parental lines for mating by looking at the relationship between genetic markers and the traits they try to predict [82] (Figure 1). These techniques support more effective and focused breeding programs by enabling accurate selection and uncovering the genetic basis for the trait of interest, ultimately improving food output and resolving global food security issues [83].

### 3.4. Germplasm Characterization

The essential step in the production of hybrid rice is the characterization and identification of parental lines in the germplasm. Phenomics and genomics methods are used to identify and describe rice germplasm accessions. Some methods, such as chemical tests [84] and haplotype characterization [85], are used to locate and validate grain shape loci and trait markers. In some of the studies that assessed the use of core set markers, the genetic makeup and population layout of wild genotype and Indian rice cultivars were analyzed to determine the genetic diversity of the rice germplasm [86], along with the genetic diversity and molecular characterization of several genotypes of *Oryza glaberrima* and *Oryza sativa*. Identifying restorer and maintainer lines, as well as analysis of agronomic parameters like plant height and maturity days, are all part of the evaluation of rice germplasm for the production of hybrid rice [87]. Furthermore, it is important to characterize and assess the rice blast resistance of the Chinese indica hybrid rice parental line to produce hybrid rice [88]. Examining the genetic variation in the floral and agronomic traits, which are crucial for enhancing hybrid seed setting in rice, is a key component of rice germplasm characterization in floral traits for hybrid rice production [34]. The properties of stigma, spikelets, and grains for each panicle, length of panicle, reproductive organs, weight of seed test, photosynthesis, seed setting rate, conductance, awn length, and transpiration are a few of the floral traits that are investigated [89]. Rice plant reproductive structures and processes must be carefully examined and understood to characterize rice germplasm for floral characteristics to the production of hybrid seeds [90]. Following are some crucial aspects of rice germplasm assessment for floral characteristics in the production of hybrid rice:(1)Analyzing the structure and morphology of spikelets are the basic parts of rice inflorescence [91].(2)Evaluate the florets’ fertility, including the number of florets that mature into fully developed seeds and the amount of floret fertility per spikelet [92].(3)Estimating the time between planting and flowering is essential for coordinating the flowering of many parental lines and enabling regulated pollination [93].(4)Determining the pollen’s viability to make sure it can successfully fertilize for hybrid rice production [94].(5)Evaluating the time when the stigma is susceptible to pollen and the amount of exsertion it receives. This information is used in the planning of controlled pollination for hybrid rice production [16]. Examining the mechanism and time of pollen release in anthers is crucial for synchronizing pollination programs [95].(6)Genetic markers, yield components, seed setting rate, and CMS lines also play an important role in the germplasm characterization of floral traits for hybrid rice production [76,96,97].

### 3.5. Role of QTLs for Hybrid Rice Production

Quantitative trait loci (QTLs) are linked to specific traits or phenotypes and their investigation can speed up the marker-assisted breeding of any crop. To choose parental lines with acceptable features for hybridization, it is critical to identify QTLs for the parental lines used in the development of hybrid rice. Identification of QTLs in rice is commonly carried out via QTL mapping [97,98]. Moreover, several QTLs have been identified by using markers for different floral traits in previous research (Table 2). QTLs can be employed to enhance floral characteristics and boost seed output in hybrid rice. For five floral traits, such as style length, stigma length, and stigma exsertion rate, a study found 14 QTLs originated from the perennial wild rice *Oryza longistaminata*. Using a BC_2_F_2_ mapping population produced from OL and IR64 as the donor and recipient parents, respectively, these QTLs were discovered by linkage analysis (Table 2). None of the QTLs were introgressed into the backgrounds of the CMS line to confirm their genetic impact and assess the out-crossing rate [77].

Another study found that qSE7 is a significant QTL affecting rice stigma exsertion rate. Several QTLs for stigma exsertion rate (SER) have been discovered on all of the rice chromosomes. Stigma exsertion is a crucial floral characteristic for the development of hybrid seeds. However, the primary obstacles to the widespread application of marker-assisted selection (MAS) for high-yield hybrid rice breeding at this time are the instability of QTL expression and the unavailability of trustworthy markers [99]. In a recent study, multiple QTLs were pyramided in an attempt to rebuild the high stigma exsertion rate (SER) in rice. The study found 18 QTLs linked to SER, and the outcomes demonstrated that rice could successfully improve the SER trait by pyramiding several QTLs [100].

**Table 2 plants-13-00578-t002:** Number of QTLs identified by using different markers for floral traits of rice in different years.

Traits	QTLs	Markers	References
Female floral traits (stigma length, style length, stigma breadth, stigma area, and pistil length)	14	164 polymorphic SSR and STS markers	[77]
Flower morphology (filament, anther length, style length, palea, lemma)	11	180 SSR markers	[101]
Stigma exertion	8	213 SSR markers	[102]
Pistil, stamen, size, and shape of glume	7, 4, 14, 6	147 markers, mostly RFLP	[103]
Stigma exertion	11	171 SSR markers	[104]
Anther length and stigma exertion	4	120 RFLP markers	[105]
Anther length	4	181 RFLP markers	[106]
Glume, pistil, and stamen	160 QTLs	182 RFLP markers	[107]

### 3.6. Contribution of GWAS towards Hybrid Rice Production

Since the development of higher genomics instruments like SNP arrays, one especially well-liked quantitative genomics technique, the genome-wide association study (GWAS) has grown in significance for the mapping of quantitative features [108,109]. GWAS has been useful in finding several key chromosomal regions that control several complex agronomic characteristics, including rice panicle length, plant height, and flowering time [110] (Figure 1). Furthermore, GWAS was used for stigma exsertion to uncover genetic loci that are strongly correlated with it, as well as other floral traits in rice. The researchers used 6.5 million single nucleotide polymorphisms from 533 different Oryza sativa populations to conduct a GWAS on floral traits. They discovered 23 genetic loci that are strongly linked to stigma exsertion and related characteristics, three of which are co-located with three key genes (GS3, GW5, and GW2). Further research revealed that these three genes influenced stigma exsertion by regulating the shape and size of stigma and spikelet. The levels of stigma exsertion and related features were mostly characterized by GW5 and GS3 combinations [111].

Genome-wide association studies (GWAS) is a relatively recent method for rice geneticists to explore the genetics of complex traits across varieties and find the causative genes or the causative loci that drive these traits. Genome-wide association studies (GWAS) used statistical approaches for links between phenotypic variation and sequence polymorphisms in rice varieties [112]. The importance of GWAS for marker-assisted breeding still has to be further validated in breeding populations despite its value in discovering novel genes for complex traits. Despite its many advantages, it has certain limitations, which include the low production of hybrid seeds, the high cost, and low power to detect loci with minor effects and heterosis complexity in hybrid rice [113]. When using GWAS for hybrid rice cultivation, limitations should be taken into account. Therefore, it has been suggested that conducting association studies using diverse germplasm is a better option because the outcomes may be immediately integrated into successful breeding efforts [114]. Besides identifying genomic regions associated with floral traits or the traits of interest, it is very important to produce easily navigable markers for rice breeding using marker assistance. Here, Kompetitive Allele-specific PCR (KASP) genotyping allows the research community to develop a user-friendly and precise marker system that can speed up the marker-assisted breeding of any crop [74].

### 3.7. Transgenic Technology for Floral Traits for Hybrid Rice Production

Transgenic technology has been successfully employed to improve floral traits in hybrid rice cultivation through genetic engineering methodology. This technology uplifts the outcrossing rate of CMS (cytoplasmic male sterility) lines and increases hybrid rice production [20]. Moreover, floral traits of CMS, including stigma exsertion rate, stigma size, anther size, style length, anther in pollen parent, filament length, and pollen number and anther in pollen parent, affect the ability of outcrossing rate that are mainly targeted to increase the hybrid rice production [20]. By enhancing these characteristics, it is possible to significantly improve the outcrossing ability of cytoplasmic male sterile lines, which can overcome the poor production as well as the high cost of hybrid seeds for large-scale production of hybrid rice [56]. Blooming time, blooming length, angle of floret opening, duration of floret opening, and the number of blooming days are the flowering behavior that also affects the outcrossing rate in rice [101]. The development of transgenic technology for floral traits for hybrid rice production can be achieved in several ways. For instance, one study sought to develop a smart sterile line for the growth of hybrid rice by using an artificial cytoplasmic male sterile lines gene as a pollen killer [19]. Another work used a transgenic construct to modify a recessive nuclear male sterile (NMS) mutant to create a weight-based seed-sorting mechanism for third-generation hybrid rice [115]. Furthermore, DuPont Pioneer has developed a seed production method for maize by introducing a fertility-restoration gene into a male sterile mutant. This method may also be used for the breeding technology of hybrid rice [116]. It is important to note that transgenic rice has been developed with several characteristics, such as high protein content, higher yield, herbicide resistance, and biotic and abiotic stress tolerance (Table 3). However, the application of transgenic technology in rice has sparked questions regarding the possibility of gene flow to close relatives that are weedy, as well as wild using pollen-mediated flow of genes [117]. Moreover, (*OsNP1*) *Oryza sativa* No Pollen 1, a nuclear gene, is used to develop a rice male sterility system employing transgenic technology for floral traits for the development of hybrid rice. For rice pollen development and male fertility, the *OsNP1* gene controls pollen exine and tapetum degeneration for the production of male fertility in rice. The researchers developed a male sterility system that may overcome the inherent issues of both CMS and PTGMS systems by converting the osnp1 mutant, with *OsNP1* paired with a gene that deactivates the gene and transgenic pollen that labels the recombinant seed. The results of the hybrid breeding test on the male sterile line produced by this method demonstrate the technique’s considerable practical potential for hybrid rice breeding and production [116].

### 3.8. Development of New Hybrid Rice Breeding Strategies

New hybrid rice breeding strategies are also being developed to address specific challenges and goals in hybrid rice production [123]. For example, the development of submergence-tolerant hybrid rice varieties has become a priority in areas prone to flooding [124] (Table 3). Another strategy involves the use of genomic selection, which uses genomic information to predict the performance of potential hybrid offspring and select desirable traits before the hybridization process begins [125].

Overall, advances in hybrid rice breeding technology have significantly improved the efficiency and effectiveness of hybrid rice production [126]. By combining traditional breeding methods with new techniques and technologies, breeders can produce higher-yielding and more resilient hybrid rice varieties to meet global food security challenges [127]. Furthermore, exploiting the CMS system [128], improving floral characters and outcrossing rates [129], enhancing stigma exsertion [111], and extensively assessing parental lines are all necessary for creating novel hybrid rice breeding techniques for floral traits. Breeders can create new hybrids and promising parental lines that can enhance hybrid rice seed sets by putting these techniques into practice [130].

## 4. Hybrid Rice Production: A Way Forward to Combat Global Food Scarcity

Hybrid rice production has come a long way since its beginnings in the 1920s [131]. During the 1960s, China emerged as the pioneer in producing the first commercially viable hybrid rice [16]. Moreover, the technique for mass production of hybrid seeds had been developed in 1975, and within a year, hybrid rice became accessible for commercial use. As a result, China proved to be the first nation to use a hybrid rice program in real meaning to produce hybrid rice on a large scale [132]. Hybrid rice was seen as a remedy for the growing population and the shortage of cultivated land in China, which put pressure on the national food economy [8]. The adoption and cultivation of hybrid rice in India, Bangladesh, China, Pakistan, Vietnam, and the Philippines has led to significant increases in rice production [133]. The use of rice hybrids has contributed to significant increases in rice production, particularly in China and other countries [134]. The success of hybrid rice is largely attributed to breakthroughs in breeding methods and the manipulation of floral traits, such as photoperiod sensitivity and temperature sensitivity [135]. To fulfill the demand for future rice, breeding technology for the super rice variety was recognized in 1996, aiming to increase rice production [8]. Since then, decades of effort and studies have resulted in novel germplasm resources, varieties, and cultivation programs and techniques, significantly contributing to rice research development and improving China’s food security issues [8].

A vital component of China’s supply of food is hybrid rice [13]. Compared to ordinary rice, hybrid rice yields are 20–30% more powerful. Hybrid rice has been produced in over 6 billion hectares in China, which has increased the nation’s cumulative rice yield by more than 0.6 billion tons. In recent years, China’s hybrid rice area has expanded to 16 million hectares annually, making up 57% of the total rice production area and almost 65% of the total yield of rice in China. Commercial hybrid rice has helped China to achieve its goal of grain self-sufficiency by increasing production by around 2.5 million tons annually, which could provide food for a further 80 million people. This also highlights China’s contribution to the advancement of agricultural technology [12]. China significantly increased the rice output due to the public commercialization of hybrid rice, dramatically improving yield per hectare and overall production [12]. There are currently more than 120 countries that grow rice on five continents. Moreover, 110 million ha of rice are grown outside of China, primarily in Asia, Africa, and America [16]. According to a report from 2014, hybrid rice occupied an area of 6.36 million hectares. Bangladesh, Pakistan, India, Indonesia, the Philippines, Myanmar, Vietnam, Sri Lanka, Iran, the United States, Brazil, Argentina, and Uruguay are among the countries developing hybrid rice [12,16]. The widespread usage of hybrid rice has been constrained due to factors such as high production costs and low acceptance among poor rice farmers in less developed countries [136]. Despite these obstacles, hybrid rice is still a crucial technology for raising rice production and supplying food security at several locations globally [17].

## 5. Floral Traits and Their Importance towards Hybrid Rice

Hybrid rice technology has revolutionized the farming practice of rice as compared to inbred rice. However, it was a key factor in developing proper field management techniques for the maximum rice yield. Many researchers apply systematic management techniques, such as QTL and precise rice methods, for high yields. Additionally, there should be an improved system for the entire world to cultivate rice, making it a significant crop. Floral traits determine the success of cross pollination as rice is a self-pollinated crop, but for hybridization, cross pollination is essential [137,138]. The critical floral traits in rice include spikelet opening angle, stigma exsertion, flowering behavior, and pollen longevity. Spikelet opening angle and stigma exsertion are the two most crucial floral traits that influence cross pollination in rice [139]. The manipulation of floral traits, such as stigma exsertion, can increase cross pollination and higher hybrid seed production [137]. Therefore, an improved understanding of floral traits is essential in developing new and improved hybrid rice varieties [128]. The utilization of these factors has contributed significantly to the development of hybrid rice cultivars with increased tolerance to biotic and abiotic stresses, better grain quality, and greater yields [81]. Moreover, flowering has numerous meanings in rice, such as initiation, structure development, flower opening, and heading [140] (Figure 2). The primary unit for the grass inflorescence is the spikelet, which has florets and glume. The floret has a pair of palea and lemma, carpel, lodicules, and stamens. Among different species of grass, the floret number varies in a spikelet. Spikelet contains a sterile lemma pair, a fertile floret, and a pair of glumes in a highly reduced form, which are rudimentary glumes (Figure 2) [141]. In addition, spikelet development in grass species regulates the grain yield and reproduction process, and the molecular mechanism behind the development of grass spikelets has gained more interest from researchers in recent years [142].

Several investigations have been carried out on the genetic and molecular factors underlying rice flowering. Recent advancements in the field of molecular genetics facilitated the researchers to better understand the roles of the quickly growing variety of genes associated with rice flower production. The rice flower genetic architecture and the genetic framework of model eudicots are similar to rice flower development. However, it is also likely that rice makes use of certain genetic pathways, which helped to build the distinctive floral architecture of rice [143]. A sophisticated gene network that incorporates environmental signals like day duration (photoperiod) and light accurately controls when flowers bloom. The Arabidopsis ABCDE genes and sets of genes encoded in rice share a lot of similarities, pointing to a shared genetic process in both plants [143]. The Hd1-dependent pathway contains Hd3a and RFT1, two important florigen genes with four regulatory modules that make up the rice flowering pattern [144]. Genetic and MutMap investigations depicted a single recessive gene on chromosome 6 for controlling the number of floral organs [145].

Recent advancements in floral traits and breeding technology have revolutionized hybrid rice production. Moreover, different floral and floral-related traits such as stigma exsertion, anther, carpel, stamen, palea, lemma, glumes, spikelets, photoperiod sensitivity, and flowering behavior play a vital role in hybrid rice production (Figure 2). These advancements have allowed researchers to produce rice cultivars with enhanced tolerance to biotic and abiotic stresses, increased production potential, and superior quality [60]. The floral traits, such as anther and stigma exsertion, have played a major role in the breeding technology of rice. These traits are discussed in detail below:Anther length

Anther is present in stamen as part of it, with pollens and anther cell layers. The defective development of anther causes pollen sterility; therefore, plant yield is also determined through the control of anther development at the reproduction level [146]. At the initial stage of meiosis in rice, microspores occur in the anther, which is covered by four somatic layers of anther wall, such as tapetum, epidermis, middle layer, and endothecium. Tapetum development is important for the process of PCD (programmed cell death) at the micro-spore level, and abnormal tapetum development causes male sterility in rice [147]. Abiotic stress can easily target pollination in rice. Every day, as the temperature rises to 35 °C, floret sterility develops at the flowering stage [106] or at the young level, when the temperature drops below 20 °C at the stage of micro-spore formation. High temperatures disturbed the release of pollens on the stigma from the anther at the flowering stage [148]. Furthermore, anther length is correlated with pollen grain numbers (Figure 2) and long anthers can overcome the shortage of normal pollen, which is caused by cool temperatures. Therefore, a key agronomic characteristic of seed setting is anther morphology and pollination stability at extreme temperatures [106].

b.Stigma exsertion

Stigma exsertion is a valuable trait to increase hybrid rice production. Numerous studies have been conducted, such as the detection and mapping of quantitative trait loci [149,150], the development of pyramiding lines [100], and the epistasis of QTL for analysis of the stigma exsertion rate in plants [151]. These studies also showed that stigma exsertion is a measurable trait, and only a main gene can control this trait [152]. The high proportion of stigma exsertion in the maternal parents overcomes the synchronization barrier of flowers between the paternal and maternal parents, and the maternal parent catches more paternal parent pollens [99] (Figure 2). Researchers consistently consider stigma exsertion due to its involvement in the high yield of hybrid seeds and male sterile production [153]. The genetic bases, especially the fertility re-establishment of stigma exsertion and cytoplasmic male sterile lines, were explained by maintainer lines that show less stigma exsertion [154]. These lines also seriously affect the hybrid seed production in rice. Cytoplasmic male sterile lines failed to produce seeds with a lower yield, possibly due to a low proportion of stigma exsertion. The commercial cytoplasmic male sterility line with a higher proportion of stigma exsertion is K17 [78].

Moreover, GS3 and grain size QTL were confirmed on chromosome 3 repeatedly overcoming the stigma length and stigma exsertion [155]. The genetic bases for stigma exsertion are still unclear and need further studies. The rate of stigma exsertion was grouped into SSE (single stigma exsertion), DSE (dual stigma exsertion), and NSE (no stigma exsertion), as illustrated in Figure 2. However, stigma exsertion was observed after 5–7 days of heading, and five panicles from every line of two different plants were selected.

Single stigma exsertion: The stigma exsertion appeared just on one side of the spikelet.Dual stigma exsertion: The stigma exsertion appeared on both sides of the spikelet.Total stigma exsertion: The stigma exsertion occurred in addition to dual and single stigma exsertion on the spikelet.No stigma exsertion: The stigma exsertion does not occur on the spikelet [104].

Previous studies showed that the F_2_ population from K17B and Huhan1B as CMS maintainers are used for the mapping of quantitative trait loci that affect rice stigma exsertion [78]. A total of 92 SSR markers were used to identify QTLs that affect the SSE (single stigma exsertion), DSE (dual stigma exsertion), and TSE (total stigma exsertion) [150]. There are two methods available for the quantification of SE, and one is the Whole Panicles Method: The categorization of single, dual, total, and no stigma exsertion is carried out by using a lens with an illuminated magnifier. In this method, each panicle’s specific spikelets pass through the process of separation. The other method is the panicles zone method: All panicles undergo the process of division into three different zones—lower, middle, and upper—for analyzing the trait of stigma exsertion. Using these three different zones, the five spikelets were randomly collected to observe stigma exsertion. The panicle zone method reduced the human resources and time as compared to the panicle’s whole method [156].

c.Photoperiod sensitivity

Floral traits in rice, such as photoperiod and temperature sensitivity, play significant roles in the development of hybrid rice [157]. Photoperiod sensitivity refers to the response of rice to the length of daylight, which affects flowering time and, ultimately, the yield of hybrid rice [158]. The manipulation of photoperiod sensitivity has been a key factor in the success of hybrid rice production, as it allows for the synchronization of male and female flowering and the production of more viable seeds [159], affecting the timing of flowering such as early flowering rice varieties can be used as female parents in hybridization to produce early maturing hybrid varieties [16] (Figure 2). Late-flowering rice varieties can be used as male parents in hybridization to extend the flowering period and increase the genetic diversity of hybrid rice [160]. The rice plants’ sensitivity to day length enables them to modify the timing of their flowering in response to seasonal fluctuations, thereby enhancing reproductive success and output [161]. In response to photoperiod, several important genes and mechanisms that regulate flowering timing have been found [162,163]. Heading date 3a (Hd3a), and Rice Flowering Locus T1 (RFT1), among others, are examples of these genes [164]. Understanding the genetic underpinnings of photoperiod sensitivity can offer insights into controlling rice flowering time, which may be significant for crop breeding and creating better cultivars [157].

## 6. Prospective Advances in Breeding Technology of Hybrid Rice

However, with the ever-increasing population and changing environmental conditions, new challenges are emerging in rice production, requiring continuous innovation in hybrid rice breeding technology [165].

### 6.1. Here Are Some Future Directions in Hybrid Rice Breeding Technology

Creating hybrid rice cultivars that possess resilience to both abiotic and biotic stresses, including diseases. Rice plants are vulnerable to a range of challenges, including pests, diseases, and unfavorable abiotic, biotic, and climatic conditions.Developing biofortified hybrid rice: Rice is a major staple food for a huge population of the world. However, it is deficient in several essential nutrients, including iron, zinc, and vitamins and these mineral elements are essential for the normal growth and development of human beings. Developing hybrid rice varieties with improved nutritional value can help to address these deficiencies and improve the health of rice consumers [166].Developing hybrid rice varieties with higher yield potential: Despite the significant yield gains achieved through hybrid rice breeding technology, there is still room for improvement. Developing hybrid rice varieties with higher yield potential can help to meet the growing demand for rice and reduce the pressure on land use. Scientists are employing progressive breeding methods, such as genome editing and CRISPR/Cas9, to identify and incorporate yield-enhancing genes [167] that confer drought, heat, and salinity tolerance into hybrid rice varieties [168].Developing hybrid rice varieties with enhanced resilience to climate change: Climate change significantly threatens rice production, affecting yield and quality. Developing hybrid rice varieties with enhanced resilience to climate change is critical for ensuring this [169].Developing hybrid rice varieties with enhanced agronomic traits: Agronomic traits, such as plant height, tillering ability, and panicle size, play a critical role in rice production. Developing hybrid rice varieties with enhanced agronomic traits can help to improve rice yield and quality. Researchers are using innovative breeding practices, such as quantitative genetics and phenomics, to identify and incorporate genes that confer desirable agronomic traits into hybrid rice varieties [56,170].Use of machine learning and artificial intelligence: The use of machine learning and artificial intelligence (AI) could help to identify patterns and relationships in large datasets, allowing for more effective prediction of hybrid rice performance [171]. In hybrid rice breeding, various parental lines are chosen and crossed to produce new lines with enhanced features by using this technique. AI makes it simple and easy to combine and analyze various datasets, offers models for the performance prediction of hybrid [172], and helps the breeders and researchers to make wise choices and increase the effectiveness, accuracy, and success rate of hybrid rice breeding. This technique ultimately increased agricultural yields and food security [173].Incorporation of genomic selection: Genomic selection could allow for the selection of desirable traits based on genetic markers, even before the phenotype is observed, leading to more efficient breeding [174].

These efforts will help to ensure sustainable rice production and contribute to global food security.

### 6.2. Potential Impact of Emerging Technologies on Hybrid Rice Production

Emerging technologies such as gene editing, artificial intelligence, and machine learning could revolutionize hybrid rice production by allowing for faster, more efficient, and more precise breeding. These technologies could help to address the challenges faced by hybrid rice production, such as the need for stress tolerance and disease resistance, while also expanding the market for hybrid rice by developing varieties with desired traits. These technologies have the potential to revolutionize the identification and exploration of key floral traits in hybrid rice production. By identifying and understanding these traits, breeders can develop hybrid rice varieties with improved yield, quality, and sustainability. Here are some potential impacts of emerging technologies in identifying and exploring key floral traits in hybrid rice production:i.Genomics and molecular breeding: Advances in genomics and molecular breeding technologies have greatly accelerated the identification of key floral traits in hybrid rice varieties [175]. These technologies can identify genes associated with specific floral traits, such [175] as flowering time, panicle size, and pollen viability, allowing for targeted breeding and genetic modification of hybrid rice varieties [176,177,178]. Moreover, the use of molecular markers in plant breeding to explore important traits has gradually increased in recent years. The foundation for developing these molecular markers is QTL mapping, which identifies the genetic loci that are quantitatively associated with desirable traits [179]. Marker-assisted selection is the targeted modification of a certain genomic area that affects how a desired characteristic expresses itself in a short time by using DNA markers. These developments have propelled molecular breeding technology into the research of innovations [180]. Plant breeding technology also used marker-assisted selection to uplift the biotic and abiotic stress tolerance, and, ultimately, this process increased crop production. Marker-assisted selection uses the linkage disequilibrium (LD) between QTLs and markers, which entails the non-random connection between QTLs’ alleles and markers [181]. Identifying the target trait’s genes and the markers closely associated with quantitative trait loci (QTLs) is crucial before using marker-assisted selection [182]. Marker-assisted selection is distributed into four categories: marker-assisted pyramiding, marker-assisted backcrossing, marker-based recurrent selection, and early-generation marker-assisted selection. These methods classify the early genetic material in different generations and strongly impact the breeding cycle [74]. Marker-assisted selection (MAS) in plant breeding has some restrictions despite its benefits. The poor selection of traits that are controlled by several minor effect alleles is one such restriction [183]. Moreover, marker-assisted selection also becomes limited when numerous genes with minor effects are present to control one trait. Therefore, this selection is optimum for qualitative characters, not quantitative traits [184]. New methods are being used to get around these restrictions. Predictive breeding techniques that use agro-big data, sophisticated statistical models, and basic machine learning algorithms have recently gained popularity as efficient methods for overcoming MAS’s limitations [185].

However, the majority of the markers that were previously identified still need to undergo validation and be transformed into reliable markers. Besides their validation, the identified markers cannot be used for the marker-assisted breeding of any crop. Lately, LGC Limited developed a Kompetitive allele-specific PCR (KASP) genotype identification test, which has become a popular tool for SNP genotyping [186]. KASP is a PCR-based homogeneous fluorescent genotyping method that has various advantages, i.e., cost-effectiveness, precision, and being trustable [187]. Numerous research studies are available on rice where KASP markers have been developed for the traits of interest [188,189,190]. However, to the best of our knowledge, no study is available claiming the development of KASP markers for floral traits in rice. A good number of QTL and GWAS studies facilitated the genomic regions associated with floral traits in rice. There is a need to develop user-friendly KASP markers using previously identified genomic regions through QTL mapping and GWAS to speed up the marker-assisted breeding of rice regarding floral traits.

Marker-assisted selection and genomic selection are used in molecular breeding technology to select plants with desired traits. GS is a novel method that can move beyond marker-assisted selection’s drawbacks (Figure 1). Furthermore, phenotypic evaluation requires multiple years to check the relationship between genotype and environment for the betterment of complicated traits. Nevertheless, it was challenging because it was time consuming and required high costs. The implementation of Next-generation Sequencing (NGS) has drastically decreased the costs involved in genome sequencing, making it easier than ever to obtain high-resolution genome data. GS has become a practical and affordable tool due to continuous advancements in sequencing methods [191].

Genomic selection (GS) in crops uses the whole range of high-performance markers in the genome [192]. GS models take into account all markers across the genome that affect a trait without applying any strict threshold criteria, in contrast to conventional MAS, which concentrates on a small number of significant genes or quantitative trait loci (QTLs). The combined effects of many markers, including minor genes or QTLs, are taken into account by the GS method by combining all accessible markers in the prediction model. This strategy prevents the genetic diversity linked to these minor characteristics from being lost. It is determined that GS is preferable to MAS for traits controlled by several markers [193]. Genotyping information from seeds or seedlings can be used to predict the phenotype of mature individuals, which is a significant benefit of genomic selection (GS). As a result, crop varietal development is significantly accelerated by eliminating the need to spend numerous years conducting prolonged phenotypic studies. Moreover, this simplified strategy boosts effectiveness by utilizing genotype data to produce precise phenotype predictions, saving time and money in the breeding process [194]. Both simple traits with high heritability and complicated ones with low heritability exhibit efficiency under genomic selection (GS) [82]. Although genomic selection (GS) is an effective method for plant breeding, developing an ideal statistical model for its use presents several difficulties. Little information is available regarding the ideal setup of these models. The effectiveness of GS can be affected by problems like faulty data imputation, environmental restrictions, and unexpected responses. Moreover, several trials have been conducted using different prediction models to solve these problems, but they still do not work well with multi-dimensional genome data [195]. A variety of models are used in the field of genomic prediction (GP) to predict phenotypes using a large number of markers, with each model based on assumptions on the variations and distributions of the markers [196]. Currently, repeating trials using various statistical models is the main approach to overcome this problem. This strategy iteratively seeks to find an optimum scenario that can be successfully applied to the desired objectives [195].

ii.High-throughput phenotyping: High-throughput phenotyping technologies, such as crewless aerial vehicles (UAVs), can quickly and accurately collect data on floral traits of hybrid rice varieties. This can help breeders identify and select hybrid rice varieties with desirable floral traits, improving yield and quality [197,198,199].iii.Imaging and machine learning: Imaging and machine learning technologies can analyze large datasets of floral traits, identifying patterns and relationships between different traits. This can help breeders understand the complex interactions between different floral traits and develop hybrid rice varieties with optimized floral architecture and reproductive biology [200].iv.Metabolomics: Metabolomics technologies can analyze the chemical composition of rice flowers, identifying the compounds that contribute to aroma, flavor, and nutritional value. This can help breeders to develop hybrid rice varieties with improved sensory and nutritional characteristics [201].v.Gene editing: CRISPR/Cas9 gene-editing technology can be used to modify genes associated with key floral traits in hybrid rice varieties precisely. This can enable breeders to develop hybrid rice varieties with optimized floral architecture and reproductive biology, improving yield and quality. To enhance desirable characters, CRISPR-Cas9 can be used to insert particular genetic alterations into parental lines [202]. It makes it possible to specifically modify the genes linked to desired traits like productivity, disease resistance, stress tolerance, and nutritional quality [203]. Breeders can increase hybrid rice production with better qualities by directly altering the DNA [126]. In rice plants, certain genes can be “knocked out” or turned off using CRISPR-Cas9. Breeders can deliberately delete genes that cause undesirable qualities or prevent the development of desired traits by using CRISPR-Cas9 and improving the hybrid rice performance [204]. Modifying floral traits can be useful in the production of hybrid rice for several reasons. For flower synchronization, CRISPR cas9 technology was used to control flowering time in plants. For aberrant floral growth or early flowering, genes including SVP, TFL1, and AP1 were addressed in Arabidopsis [205]. Other floral properties can also be altered using CRISPR besides flowering time. To produce pale blue flowers, the CRISPR/Cas9 system has been successfully applied to alter the color of flowers in *Torenia fournieri* [206], as well as multiplex CRISPR-Cas9, which introduce novel flowering and architectural characteristics in hexaploid *Camelina sativa* [207]. For instance, adjusting the flowering period can assist in synchronizing the female and male parental lines’ flowering, which is important to increase hybrid rice production [37]. Researchers often identify the genes responsible for the qualities they want to modify before using gene editing to change floral attributes in hybrid rice [60]. Next, they create a guide RNA (gRNA) specially designed for the target gene and insert it into the rice plant cells with the CRISPR-Cas9 system. The Cas9 protein makes a double-stranded break in the DNA of the target gene after being directed to it by the gRNA. After the break, the plant’s natural DNA repair system repairs gene mutations or insertions/deletions (InDels) that can change floral traits [208], as shown in Figure 1. However, CRISPPR Cas9 also has some limits, such as difficulty in obtaining higher editing efficiency, the transfer of technology to different rice populations, in vitro transcription, off-target effects, and plant lethality, as well as the use of common Cas9, can limit the editable range in plants, despite the obstacles, and CRISPR Cas9 proved useful in hybrid seed production [209,210,211]. Researchers can modify numerous genes at once using CRISPR-Cas9 and other gene editing methods, enabling the fine tuning of floral traits in the development of hybrid rice [203].

By leveraging these technologies, breeders can develop hybrid rice varieties with improved yield, quality, and sustainability, ensuring a secure and stable food supply for the growing global population.

## 7. Conclusions

The role of floral traits in hybrid rice production is crucial for the development of high-yielding and sustainable hybrid rice varieties. Breeding technology has made significant progress in identifying and understanding the key floral traits and developing new hybrid rice varieties with desirable traits. However, challenges such as genetic complexity and environmental factors still need to be addressed. The future of hybrid rice production lies in the development of more efficient and sustainable breeding strategies focused on floral traits, as well as the application of emerging technologies such as genomics, precision agriculture, robotics, and machine learning. By the adoption of these technologies and strategies, breeders can continue to improve the yield, quality, and sustainability of hybrid rice production, ensuring a stable food supply for the growing global population.

## Figures and Tables

**Figure 1 plants-13-00578-f001:**
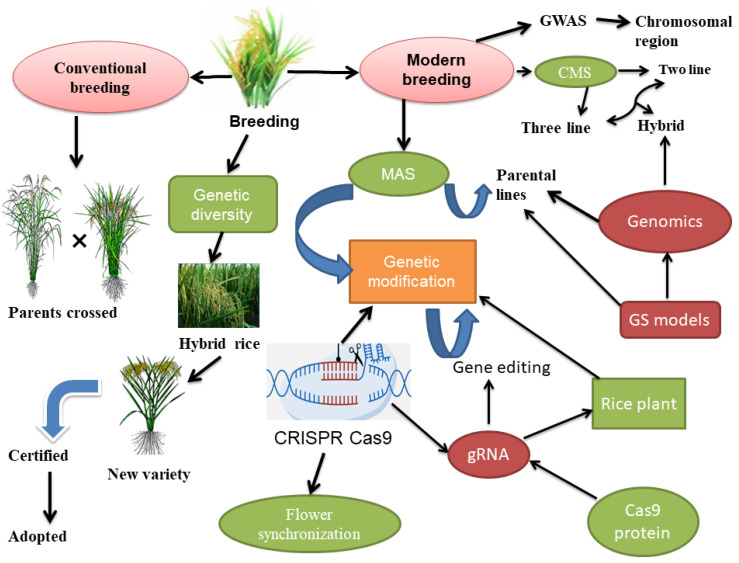
Schematic illustration of conventional and modern breeding strategies for hybrid rice production. GS (Genomic Selection), CMS (Cytoplasmic Male Sterile lines), GWAS (Genome-wide Association Study), gRNA (guide RNA), CRISPR (Clustered Regularly Interspaced Short Palindromic Repeats), MAS (Marker-assisted Selection).

**Figure 2 plants-13-00578-f002:**
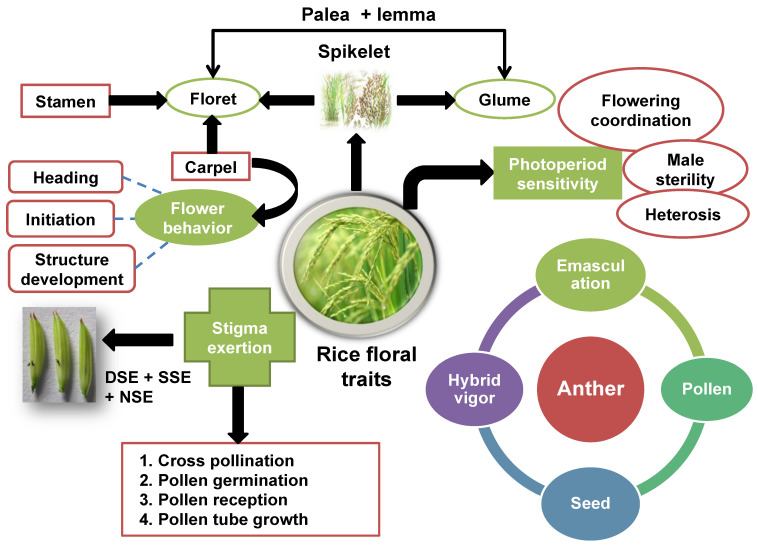
Rice floral and their related traits and their functions in hybrid rice production. DSE (double stigma exertion), SSE (single stigma exertion), NSE (no stigma exertion).

**Table 1 plants-13-00578-t001:** Hybrid rice cultivars and their developing organizations in different countries.

Hybrid Rice Cultivars	Developing Organization	Country	Reference
117 rice hybrid varieties	Public and private sector	Brazil, the United States, Egypt, India, Bangladesh, China, Vietnam, the Philippines, Indonesia, Myanmar, and Sri Lanka	[29]
Two-line, three-line, and super-hybrid rice	IRRI, China, and India	China	[6]
Local rice hybrids	The Global Rice Research Institute and Egypt’s rice research program	Egypt	[30]
50 high-yielding hybrid lines	African programme for rice	Several countries in Africa	[31]
Three *indica* hybrid rice varieties	China National seed group Co., Ltd.	China	[32]

**Table 3 plants-13-00578-t003:** Different transgenic rice and their functions.

Transgenic Rice	Functions	Reference
Bt rice	Insect resistance	[118]
Golden rice	Increased provitamin A content	[119]
Sub1 rice	Flooding tolerance	[120]
C4 rice	Increased photosynthetic efficiency and yield	[121]
Salt-tolerance rice varieties	Tolerance towards high-salinity environment	[122]

## Data Availability

All data supporting the findings of this study are available in the paper.

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
