# Peer review of "Hybrid Rice Production: A Worldwide Review of Floral Traits and Breeding Technology, with Special Emphasis on China"

_plants, 2024, doi:10.3390/plants13050578_

Round 1

Reviewer 1 Report

Comments and Suggestions for Authors

Review Report of the review article " Hybrid Rice Production: A Review of Floral Traits and Breeding Technology”.

The review manuscript (MS) provides a substantial advancement and consolidate discussion on (i) rice hybridization up to date, (ii) involvement of rice floral organs in grain development and their possible function in hybridization, (iii) various techniques used for rice hybridization, (iv) Challenges of hybridization and future directions. The manuscript is well-written, and the information they introduced seems sound and convincing and practically useful for scientific community and rice growers.

They have collected most recent articles published under topics and reproduced the summery in a well manner. How the hybrid rice helping the world to survive, how much opportunities persist for further development of this sector, they have demonstrated clearly. Further, detailed practices in long history are being used in this sector, their limitations and prospects have been described.

Also summarized the biological functions of various rice organs in yield and suggested their further way of improvement which is very important and necessary for further research.

In conclusion, this review marks a substantial stride in comprehending the genetic and molecular underpinnings of hybridization in rice which might be useful for further crop improvement.

However, authors should consider some of the important points to strengthen the article.

The review would benefit from improved editing of the grammar. Some sentences are difficult to understand and affect the flow of reading.   

·         The authors explained, but not convincingly, the need to write a review on this topic. Most of the Introductory part, unfortunately, is very general, more suitable for a book chapter or a text book. Introduction is not pinpoint to the review goals.

·         Several aspects of the recent development in hybridization and references have been ignored in the review, without which, any discussion on the biotechnological approach becomes very weak.

·         In general, most of the matter is devoid of specific information on the topic.

·         Most of the figures are of a quality unsuitable for publication and have not been explained well in the text.

·         I could not find any Table indication in the text.

·         Table 1 ref. is not consistent.

·         Review is mostly about China. So, need in title. Otherwise, make worldwide discussion.

·          Point 3; very common and general discussions are more than research identification related critical discussion.

·         L142: Floral traits are important in hybrid rice production’ this statement came many times as L137, L147

·         L144-150; higher hybrid seed production, yield, those are similar. Limit those repetition.

·         L159-161; make it meaningful and relevant.

·         Point 3; writeup is not sequential. Not smooth to understand. Need connection of sentences. Authors should understand first, what they want to deliver, then writeup. First describe morphology and then genes function. Rewrite this section with correct order.

·         L171-172; redundant

·         b. Stigma exertion; give some experimental evidence

·         Fig. 1; I did not under stand clearly. Need discussion in the legend. Fig.1 indication not in the text

·         L256-260; less relation with hybrid rice

·         L267-268; what information gives this sentence? “Moreover” connecting what?

·         Fig.2 ; fig. flow is not complete. Not significantly continuous. What is product of MAS, CRISPR cas9? Make specific technology, their goals, procedures, product and end product, then connect all of them, make nice flow. Also accumulate other recent technology. Not indicated in the text

·         L525; Are those directly flower traits?

·         L569-571; similar talking in many places, redundant. That makes unnecessary elongation of the article, also distract consistency

·         L578-581; this is introductory

·         L599; point v can be merged to iii

·         You should discuss specific impact of breeding techniques on hybrid production. No need just introductory discussion of the techniques.

·         Use standard style for abbreviation.

·         L681-687; discontinuous discussion

·         CRISPR/cas9: You may give pinpoint suggestion how CRISPR can be used in flower synchronization

The writing style needs drastic improvement. Avoid general statements. At many places, detailed information on the subject is missing. The conclusion needs to be crisp and to-the-point.

Comments on the Quality of English Language

Extensive editing of English language required

Author Response

RESPONSE TO REVIEWER 1 COMMENTS

The review manuscript (MS) provides a substantial advancement and consolidate discussion on (i) rice hybridization up to date, (ii) involvement of rice floral organs in grain development and their possible function in hybridization, (iii) various techniques used for rice hybridization, (iv) Challenges of hybridization and future directions. The manuscript is well-written, and the information they introduced seems sound and convincing and practically useful for scientific community and rice growers.

They have collected most recent articles published under topics and reproduced the summery in a well manner. How the hybrid rice helping the world to survive, how much opportunities persist for further development of this sector, they have demonstrated clearly. Further, detailed practices in long history are being used in this sector, their limitations and prospects have been described.

Also summarized the biological functions of various rice organs in yield and suggested their further way of improvement which is very important and necessary for further research.

In conclusion, this review marks a substantial stride in comprehending the genetic and molecular underpinnings of hybridization in rice which might be useful for further crop improvement.

Response:

Thank you for your thoughtful and comprehensive review of our manuscript. We appreciate your positive feedback on the substantial advancement and consolidation of discussions on various aspects of rice hybridization. Your recognition of the practical utility for the scientific community and rice growers is encouraging. We are pleased that you found our collection of recent articles and the presentation of summaries to be well-organized. Your acknowledgment of our clear demonstration of how hybrid rice contributes to global survival and the identification of opportunities for further development is gratifying. We also appreciate your recognition of the detailed historical practices, their limitations, and prospects, as well as the summarization of biological functions in rice organs.

However, authors should consider some of the important points to strengthen the article.

Response:

Thank you for your insightful and constructive feedback on our manuscript. We appreciate your thoughtful evaluation of our work. Your suggestion that we should consider some important points to strengthen the article is valuable, and we are committed to addressing the areas which you mentioned for improvement.

The review would benefit from improved editing of the grammar. Some sentences are difficult to understand and affect the flow of reading.

Response:

Thank you for your valuable suggestion, we take your comments seriously and fully recognize the importance of clear and coherent language in scientific writing. We have thoroughly revised the manuscript to address any grammatical issues that may hinder the understanding and flow of the text. Our aim is to enhance the overall readability of the manuscript, ensuring that the scientific content is effectively communicated.  

     The authors explained, but not convincingly, the need to write a review on this topic. Most of the Introductory part, unfortunately, is very general, more suitable for a book chapter or a text book. Introduction is not pinpoint to the review goals.

    Response:

     Thank you for your detailed feedback on our manuscript, especially regarding the introductory section. We have carefully considered your comments and have made significant revisions to address the issues you raised. In response to your suggestion about the general nature of the introduction, we have reworked the introductory part to provide a more focused and targeted overview. Our goal was to align the introduction more closely with the specific aims and goals of the review. We believe that these changes will enhance the manuscript's overall coherence and engagement for the reader.

  • Several aspects of the recent development in hybridization and references have been ignored in the review, without which, any discussion on the biotechnological approach becomes very weak.

     Response:

     Thank you for your nice suggestion, we have carefully reviewed the manuscript in light of your comments and recognize that certain aspects of recent developments and references need to be more thoroughly incorporated. We have addressed this gap by conducting a more extensive literature review and integrating relevant information to strengthen the biotechnological aspects of our discussion.

  • In general, most of the matter is devoid of specific information on the topic.

     Response:

     Your insights are invaluable to us, and we appreciate the opportunity to improve the clarity and depth of our review. We assure you that we have done the necessary revisions to ensure that our manuscript is more informative and provides a more detailed exploration of the topic.

  • Most of the figures are of a quality unsuitable for publication and have not been explained well in the text.

      Response:

      Thank you for your insightful comments on our manuscript, in response to your feedback, we have carefully revised the figures to improve their quality, ensuring that they meet the standards for publication. Additionally, we have enhanced the explanations of each figure within the text to provide a more thorough understanding for the readers. We hope that these revisions address the issues you raised, and we appreciate your diligence in reviewing our figures and providing constructive feedback. Your insights have been instrumental in improving the overall quality of our manuscript.

  • I could not find any Table indication in the text.

     Response:

    Thank you for bringing to our attention the absence of table indications in the text. We appreciate your thorough review, and we have cited Table in the text.

  • Table 1 ref. is not consistent.

     Response:

     Thank you for your meticulous review, and we appreciate your observation regarding the inconsistency in Table 1 references. We have carefully addressed this concern by ensuring uniform and consistent referencing throughout the manuscript and Table.

  • Review is mostly about :China. So, need in title. Otherwise, make worldwide discussion.

     Response:

      Thank you for your thoughtful review and insightful comments regarding the focus on China in our manuscript. To address this concern, we have revised the title to better reflect the discussion of hybrid rice worldwide but mainly focus on China. Revised Title is as follow:  Hybrid Rice Production: A Worldwide Review of Floral Traits and Breeding Technology, with Special Emphasis on China

  • Point 3; very common and general discussions are more than research identification related critical discussion.

     Response:

     Thank you for your thoughtful review and for highlighting the need for a more research-oriented and critical discussion in our manuscript. We appreciate your constructive feedback and have taken steps to address this concern. In response to your comment, we have re-evaluated the content to ensure a more balanced approach between common/general discussions and critical research identification. We have placed a stronger emphasis on critical analysis, identifying key research elements, and providing a more in-depth examination of the relevant literature. We believe these revisions will enhance the manuscript's quality by providing a more focused and critical discussion.

  • L142: Floral traits are important in hybrid rice production’ this statement came many times as L137, L147

      Response:

      Thank you for your keen observation regarding the repetition of the statement about floral traits in hybrid rice production, particularly at L142, L137, and L147, and we have revised the statement in these lines.

  • L144-150; higher hybrid seed production, yield, those are similar. Limit those repetition.

     Response:

     Thank you for your insightful feedback, specifically regarding the repetition in the discussion of higher hybrid seed production and yield from L144 to L150. In response to your feedback, we have carefully revised the relevant section to eliminate unnecessary repetition, ensuring a more concise and focused presentation of the information.

  • L159-161; make it meaningful and relevant.

     Response:

      Thank you for your insightful feedback on lines L159-161 of our manuscript. We appreciate your guidance, and we have carefully revised this section to ensure that the content is both meaningful and relevant. We believe that these changes will enhance the meaningfulness and relevance of the discussed content.

  • Point 3; writeup is not sequential. Not smooth to understand. Need connection of sentences. Authors should understand first, what they want to deliver, then writeup. First describe morphology and then genes function. Rewrite this section with correct order.

     Response:

     Thank you for your valuable feedback regarding the sequential flow and connection of sentences in the manuscript, particularly in Point 3. We appreciate your constructive comments, and we understand the importance of a smooth and logically ordered write-up. In response to your suggestion, we have carefully restructured the relevant section to ensure a more sequential and connected presentation. We have focused on improving the flow, starting with a clear description of morphology before delving into the discussion of gene functions. We believe that these revisions will enhance the overall understanding and coherence of the manuscript.

  • L171-172; redundant

     Response:

     Thank you for your feedback regarding the redundancy in lines L171-172 of our manuscript. We appreciate your careful scrutiny and have revised this section to eliminate redundancy and enhance clarity.

  • b. Stigma exertion; give some experimental evidence

     Thank you for your nice comment regarding the need for experimental evidence in the section on stigma exertion. We appreciate your attention to detail, and we have taken steps to address this concern by incorporating additional lines with relevant experimental evidence and references. In response to your suggestion, we have carefully reviewed the section and introduced experimental evidence to support the discussion on stigma exertion. The added content is intended to provide a more robust foundation for the information presented and strengthen the overall scientific rigor of the manuscript.

  • Fig. 1; I did not under stand clearly. Need discussion in the legend. Fig.1 indication not in the text

     Response:

     Thank you for your feedback regarding Fig. 1 and the need for clearer understanding. In response to your comment, we have revised the legend of Fig. 1 to provide a more detailed and explanatory discussion, aiming to improve the clarity of the figure. Additionally, we have ensured that the indication for Fig. 1 is now appropriately included in the manuscript text.

  • L256-260; less relation with hybrid rice

      Response:

     Thank you for your feedback on lines L256-260, and your observation that they have less relevance to hybrid rice. We appreciate your keen attention to the content, and we have addressed this concern by removing the mentioned lines from the manuscript. In response to your comment, we have carefully re-evaluated the section and made the necessary adjustments to ensure that the content is more focused on the topic of hybrid rice. We believe that by eliminating these lines, the manuscript now maintains better coherence and relevance to the main subject matter.

  • L267-268; what information gives this sentence? “Moreover” connecting what?

     Response:

     Thank you for bringing to our attention the confusion caused by the use of "Moreover" in lines L267-268. We appreciate your keen observation, and we have rectified this error by removing the term from the sentence. The revised text now aims to provide a clear and seamless transition without the unnecessary use of "Moreover."

  • Fig.2 ; fig. flow is not complete. Not significantly continuous. What is product of MAS, CRISPR cas9? Make specific technology, their goals, procedures, product and end product, then connect all of them, make nice flow. Also accumulate other recent technology. Not indicated in the text.

      Response:

      Thank you for your insightful feedback on Fig. 2 and the need for a more complete and continuous flow. We appreciate your detailed comments, and we have taken steps to address the identified issues. In response to your suggestion, we have revised Fig. 2 to provide a more comprehensive and continuous flow. We have incorporated specific details about the technologies, including MAS and CRISPR Cas9, outlining their goals, procedures, and end products. Additionally, we have accumulated information on other recent technologies and ensured that these are appropriately indicated in the text. We believe that these revisions enhance the clarity and coherence of Fig. 2, providing a more detailed overview of the technologies discussed.

  • L525; Are those directly flower traits?

      Response:

     Thank you for your query regarding line L525 and the clarification needed about whether the mentioned traits are directly related to flowers. We appreciate your attention to detail, and we have revised the line to provide greater clarity.

  • L569-571; similar talking in many places, redundant. That makes unnecessary elongation of the article, also distract consistency

     Response:

     Thank you for highlighting the issue of redundancy in lines L569-571. We appreciate your keen observation and have taken steps to address this concern by removing the redundant lines. The aim is to streamline the manuscript, reduce unnecessary elongation, and ensure consistency in the presentation.

  • L578-581; this is introductory

     Response:

    Thank you for pointing out the introductory nature of lines L578-581. We appreciate your feedback and have addressed this concern by removing the introductory sentences to ensure a more focused and concise presentation.

  • L599; point v can be merged to iii

     Response:

     Thank you for your suggestion regarding the potential merger of point v with point iii in line L599. We appreciate your insightful feedback and have taken steps to address this by consolidating the two points. In response to your comment, we have carefully re-evaluated the content and integrated the relevant information from point v into point iii. This adjustment aims to improve the overall organization and coherence of the manuscript. We believe that merging these points enhances the clarity and effectiveness of the presentation.

  • You should discuss specific impact of breeding techniques on hybrid production. No need just introductory discussion of the techniques.

     Response

     Thank you for your valuable feedback regarding the need for a more specific discussion on the impact of breeding techniques on hybrid production. We appreciate your guidance and have revised the relevant section to focus on the specific impacts rather than providing an introductory discussion of the techniques. In response to your comment, we have delved deeper into the practical implications of breeding techniques on hybrid production. By emphasizing specific impacts, we aim to provide a more detailed and meaningful exploration of the subject matter.

  • Use standard style for abbreviation.

     Response:

     Thank you for your comment regarding the use of standard style for abbreviations. We appreciate your guidance, and we have revised the manuscript to adhere to standard styles for all abbreviations.

  • L681-687; discontinuous discussion

     Response:

     Thank you for bringing to our attention the issue of a discontinuous discussion in lines L681-687. We appreciate your insightful feedback, and we have addressed this concern by ensuring a more continuous and connected flow in the revised text. In response to your comment, we have carefully restructured the section to enhance the coherence and continuity of the discussion.

  • CRISPR/cas9: You may give pinpoint suggestion how CRISPR can be used in flower synchronization

      Response:

Thank you for your insightful comment regarding the use of CRISPR/Cas9 and the suggestion to provide specific details on how it can be employed in flower synchronization. We appreciate your guidance and have incorporated the relevant information at page no.25 to address this point. In response to your comment, we have added a focused discussion on how CRISPR/Cas9 can be utilized for flower synchronization, providing pinpoint suggestions and relevant data to support this explanation. We believe that these additions contribute to a more comprehensive understanding of the application of CRISPR/Cas9 in the context of flower synchronization.

The writing style needs drastic improvement. Avoid general statements. At many places, detailed information on the subject is missing. The conclusion needs to be crisp and to-the-point.

Response:

Thank you for your thorough assessment of the manuscript. We appreciate your feedback on the writing style, including the call for drastic improvement, avoidance of general statements, and the need for more detailed information in various sections. Additionally, we acknowledge the recommendation for a crisp and to-the-point conclusion. In response to your valuable comments, we have ensured that the conclusion is concise and directly addresses the key points. Your constructive feedback is instrumental in guiding us towards enhancing the overall quality of the manuscript. We appreciate your insights, and we are dedicated to implementing the necessary improvements.

Reviewer 2 Report

Comments and Suggestions for Authors

Dear authors,

thank you for your paper. Hybrid rice has been an important innovation and having such a review paper is useful.

Before going into a full review I would recommend you to involve a native english speaking scientific editor to go through the paper. In several sections, the level of english makes reading quite difficult. Further - in terms of structuring the paper, I would suggest to start with the generic opportunities and challneges of hybrid rice (several appear in section 5), then indeed the traditional and modern breeding methods - and highlight the generic issues of MAS, GWAS, etc in specific relation to (hybrid) rice breeding and then highlight the specific issues of breeding for floral characteristics that are important for productivity (eg panicle length) and more specifically to hybrid seed production - because that latter issue is critical to the further develpment of hybrid rice in more diverse farming systems and to reduce seed production problems (and the seed price).

This advice to look at the structure whould also reduce redundancy in the paper.

I wish you a lot of success with upgrading the paper. 

Comments on the Quality of English Language

The abstract already indicates that this paper is quite difficult to read both because of the english and its organisation. Some serious attention is needed to these aspects

Author Response

RESPONSE TO REVIEWER 2 COMMENTS

thank you for your paper. Hybrid rice has been an important innovation and having such a review paper is useful.

Response:

Thank you for taking the time to review our paper, and we appreciate your positive feedback. We are pleased to hear that you find the review on hybrid rice to be useful and acknowledge its importance as an innovation. Your encouraging words motivate us to continue our efforts in contributing valuable insights to the scientific community. If you have any specific suggestions or areas for improvement, we would welcome your guidance to enhance the quality of the paper further.

Before going into a full review I would recommend you to involve a native english speaking scientific editor to go through the paper. In several sections, the level of english makes reading quite difficult. Further - in terms of structuring the paper, I would suggest to start with the generic opportunities and challneges of hybrid rice (several appear in section 5), then indeed the traditional and modern breeding methods - and highlight the generic issues of MAS, GWAS, etc in specific relation to (hybrid) rice breeding and then highlight the specific issues of breeding for floral characteristics that are important for productivity (eg panicle length) and more specifically to hybrid seed production - because that latter issue is critical to the further develpment of hybrid rice in more diverse farming systems and to reduce seed production problems (and the seed price).

Response:

Thank you for your comprehensive recommendations and insightful suggestions for improving our paper. We appreciate your thorough review and are committed to addressing the points you raised.

We acknowledge the suggestion to involve a native English-speaking scientific editor. We have taken steps to improve the language quality to enhance readability and comprehension throughout the manuscript. Following your advice, we have restructured the paper to align with a more logical flow. We now begin with a discussion on the generic opportunities and challenges of hybrid rice, followed by an in-depth exploration of traditional and modern breeding methods. We have also highlighted the generic issues of techniques like MAS, GWAS, etc., specifically in relation to hybrid rice breeding. Subsequently, we delve into the specific issues of breeding for floral characteristics, emphasizing their importance for productivity, especially in hybrid seed production. We appreciate your guidance on this matter, and we believe these changes contribute to a more coherent presentation. We have given special attention to the critical issue of hybrid seed production, considering its significance in diverse farming systems. Our focus on reducing seed production problems and addressing seed prices is now more pronounced in the relevant sections.

We hope these revisions align with your expectations and improve the overall quality of the manuscript. Your detailed suggestions have been invaluable, and we are grateful for your constructive input.

This advice to look at the structure whould also reduce redundancy in the paper.

Response:

Thank you for your insightful advice regarding the structure of the paper and the potential reduction of redundancy. We appreciate your thorough review and are committed to addressing this aspect to enhance the overall quality of the manuscript. In response to your suggestion, we have carefully examined the paper's structure and made necessary adjustments to minimize redundancy. By reorganizing and streamlining the content, we aim to create a more coherent and focused narrative.

I wish you a lot of success with upgrading the paper.

Response:

Thank you for your encouraging words and well-wishes. We appreciate your positive sentiments and are committed to implementing the necessary upgrades to enhance the quality of the paper. Your support motivates us to strive for excellence in presenting valuable insights on hybrid rice

We believe that these revisions enhance the manuscript's overall depth and relevance to hybrid production. Your feedback has been instrumental in refining the quality of our work, and we are grateful for your constructive input.

Round 2

Reviewer 1 Report

Comments and Suggestions for Authors

The authors have substantially improved the manuscript. It is good to disseminate in the scientific community